# Destructive and Non-Destructive Evaluation of Anthocyanin Content and Quality Attributes in Red Kiwifruit Subjected to Plant Spray Treatment with Cis-3-Hexenyl Butyrate

**DOI:** 10.3390/foods14030480

**Published:** 2025-02-02

**Authors:** Micaela Lembo, Vanessa Eramo, Riccardo Riggi, Roberto Forniti, Andrea Bellincontro, Rinaldo Botondi

**Affiliations:** Department for Innovation in Biological, Agro-Food and Forest Systems, University of Tuscia, 01100 Viterbo, Italy; micaela.lembo@unitus.it (M.L.); vanessa.eramo@unitus.it (V.E.); riccardo.riggi@unitus.it (R.R.); forniti@unitus.it (R.F.); bellin@unitus.it (A.B.)

**Keywords:** red kiwifruit, NIR spectroscopy, chemometry, cis-3-hexenyl butyrate, postharvest storage

## Abstract

This work evaluated red kiwifruit plants’ spray treatment with cis-3-hexenyl butyrate (HB) as an inductor of some metabolic mechanisms related to fruit ripening, including an increase in anthocyanin content and the red hue color parameter. Considering their key role as ripening parameters for postharvest fruit quality and sorting assessment, the soluble solid content (SSC) and the flesh firmness penetrometer (FFP) were also measured. Treated plants received an application of 50 mM HB, administered exactly 2 and 4 weeks before the commercial harvest. At harvest time and during postharvest fruit ripening, near-infrared (NIR) spectral acquisitions were performed in order to check the feasibility of a rapid and non-destructive prediction of fruit anthocyanin content and SSC, coupled to destructive measurements and chemometric modelling. Regarding technological and chemical results, HB treatment indicates an optimum overall qualitative storage at 30 days. The fruit from treated plants is characterized by good quality parameters, including higher SSC, enhanced red hue (a* value) and increased anthocyanin content, despite similar weight loss to the untreated fruit. The obtained chemometric results underscore the promise and feasibility of NIRs in terms of detecting and estimating anthocyanin content and SSC in red kiwifruit, in order to pursue an evident perspective of improvement.

## 1. Introduction

Kiwifruit (*Actinidia chinensis*) plays a crucial role in the Italian agricultural context, serving as a key export commodity and gaining popularity among consumers due to its rich nutritional profile. Its relevant content of vitamins, antioxidants, and bioactive compounds contributes to a balanced diet and enhances consumer demand for this fruit in both local and international markets [1]. Globally, the leading kiwifruit producers are China (52.4%), New Zealand (13.3%) and Italy (11.5%) [2].

In recent years, consumer demand for red kiwifruit has grown, driven by its unique flavor and brightly colored pericarp [3]. The color of the fruit’s flesh has become an important focus, one which has led to the market introduction of *A. chinensis* cv “Hongyang” (cv “Red Sun”), which in turn has become the first red-fleshed kiwifruit cultivar to be grown on a commercial scale, and efforts to develop additional red-fleshed varieties are ongoing. Despite the growing interest in red-fleshed kiwifruit, little is reported in literature.

The pigmentation of kiwifruit carries significant commercial value, influencing consumer choices. These pigments contribute to their appearance and strongly appeal to consumers, leading to considerable interest in new kiwifruit varieties with red flesh [4].

In this context, taste, commercial value, and nutritional quality of fruit are typically correlated with total soluble solids content (SSC), key metabolites, and physical attributes like size/weight and firmness. SSC determines fruit flavor, while bioactive compounds, such as flavonoids, enhance human health. Size, weight and firmness also influence consumer preferences [5,6,7]. SSC is crucial for ripening, with acceptable levels at 12 to 15 °Brix for Hayward kiwifruit and 13 to 16 °Brix for Soreli kiwifruit [6,8,9,10]. Although research on red kiwifruit SSC is limited, it is already known that higher SSC correlates with a more intense red color, reaching up to 19–20 °Brix in red-fleshed varieties [11].

Anthocyanins, part of the flavonoid family, are water-soluble pigments found in vacuoles and the primary pigments responsible for the coloration of red fruit and have been gaining significant scientific interest for several reasons [12]. They influence the organoleptic characteristics of food, potentially affecting the ease of processing, and are linked to human health benefits due to antioxidant and anti-inflammatory properties [13,14]. These compounds are affected by environmental factors, agricultural practices and postharvest processing, making them relevant for maintaining the fruit’s appeal and marketability [3,15]. Enhancing anthocyanin content through biotechnology and controlled processing can improve both the nutritional and sensory qualities of food products.

Understanding the factors influencing ripening and pigmentation in red kiwifruit is essential not only for enhancing consumer appeal but also for ensuring consistent quality in the market. The conventional method used to evaluate flesh color is destructive and leads to the loss of consumable products and potential income reduction. The integration of near-infrared spectroscopy (NIR) provides valuable insights into key quality attributes, enabling growers and producers to make informed decisions about harvest timing and processing techniques. Vibrational spectroscopy in the NIR range is arguably the most widely used technique for assessing the internal quality of kiwifruit [16]. NIR spectroscopy has been validated as a non-destructive approach for assessing SSC in Thai kiwifruit, as well as for evaluating flesh color in two commercial kiwifruit varieties: green (*Actinidia deliciosa ‘Bruno’*) and yellow (*A. chinensis ‘Yellow Joy’*), aiding in the understanding of color development during ripening [17]. This is a rapid technique that requires minimal sample preparation and has proven effective for validating key quality attributes of the fruit without compromising its integrity. It offers the advantage of being easy to use and can be effectively combined with traditional analytical techniques for qualitative and quantitative analysis. When combined with multivariate data analysis, NIR can correlate spectral responses with chemical composition [18], allowing for the prediction of total anthocyanins, SSC and ripeness evolution. Near infrared acoustic optically tunable filter (NIR-AOTF) spectroscopy further enhances the non-destructive measurement of bioactive compounds, like polyphenols and anthocyanins, as already demonstrated in wine grapes [19]. The assessment that the NIR-AOTF makes regarding significant quality and commercial attributes like sugars and weight loss (WL) is also significant. It can evaluate the optimal harvest time based on ripeness [20] or monitor postharvest processes [21].

Currently, the red kiwifruit on the market often exhibits poor flesh coloration, mostly limited to the central area of the fruit, with a limited radial spread. Several studies have indicated the potential use of the cis-3-hexenyl butyrate (HB) reagent, which, when vaporized on various horticultural plants, can mediate stomatal closure, induce resistance mechanisms in response to environmental stress and enhance fruit ripening, and can trigger an increased anthocyanin content and an improved red pigmentation in fruit, such as observed in red grapes [22,23,24,25].

In light of this, our study aims to investigate the potential of this natural organic compound found in the aromatic profile of strawberries in order to enhance the ripening process of red kiwifruit by promoting anthocyanin accumulation, thereby increasing the product’s appeal. Additionally, we evaluate the effectiveness of NIR spectroscopy as a non-destructive method in comparison to traditional destructive techniques, allowing for real-time monitoring of the ripening process without compromising the fruit’s quality.

## 2. Materials and Methods

### 2.1. Experimental Design

An experimental plot of red kiwifruit vineyards at the “Tre Colli Srl” farm, located in Velletri (Rome, Italy), was selected to study the impact of HB on the postharvest fruit quality parameters during ripening. The red kiwifruit vines from a cultivar designated BAG1 (yet to be deposited in the catalogue) were in their third year of age. The field test was conducted in the early morning hours (starting from 07:00 a.m.) on red kiwifruit plants spaced 4.0 m between rows and 3.0 m apart. Two applications of HB, diluted in distilled water at a concentration of 50 mM, were carried out using a commercial shoulder sprayer. Eight treated plants (T) and eight control plants (C) were selected on a sample basis within the row. Overall, 1000 fruit were used for postharvest study (500 fruit for each of the experimental samples). Given that the commercial harvest was carried out on 23 October 2023, the first application (application A) was made 4 weeks before commercial harvesting, while the subsequent application (application B) was performed 2 weeks before commercial harvesting. The experimental trial, including the red kiwifruit plants, the sprayer for HB treatments, and the fruit under storage conditions, is illustrated in Figure 1.

### 2.2. Kiwifruit Storage Conditions

After harvesting, the fruit—the average fruit weight at harvest was between 100–130 g with no significant variation between the two treatments—were transported to the Department for Innovation in Biological, Agro-Food and Forest Systems (DIBAF) laboratory (University of Tuscia, Viterbo, Italy), where they were selected for uniformity of size, appearance and absence of defects and diseases. Then, these fruits were packed in single-layer trays and, after 24 h of storage at room temperature (20 °C ± 2 °C) (curing procedure), were cooled to 1 ± 0.5 °C with a relative humidity (RH) of 85% ± 5% in a normal atmosphere, using an ethylene absorber in the storage rooms. All analyzes were conducted at each sampling time, including harvest time (T0) and interval time of 15 days during cold storage: T1 (15 days), T2 (30 days), T3 (45 days) and T4 (60 days).

### 2.3. Physical and Chemical Parameters

At T0 and during each sampling time, twenty-five untreated fruits and twenty-five treated fruits were weighed using a digital balance (Adam Equipment Co., Ltd., Milton Keynes, UK) to monitor WL (%) during cold storage, and twenty-five fruits for each sample type were analyzed to evaluate the technological parameters (SSC, flesh color, and flesh firmness) over time. Anthocyanin content was analyzed in triplicate for each sample type at each sampling time considered.

The SSC of the fresh kiwifruit juice was measured using a digital refractometer (ATAGO, Palette PR-32, Tokyo, Japan) and expressed as °Brix.

Flesh color was evaluated on peeled fruit using a Minolta colorimeter (Minolta C2500; Konica Minolta, Ramsey, NY, USA) to assess the chromaticity values via L* (Lightness) and a* (green to red) parameters. The hue angle (h*) was calculated from a* and b* values, as reported by Mcguire [26].

A flesh firmness penetrometer (FFP) test (Mod. 53205; TR Turoni snc, Forlì, Italy) was assessed by a destructive method by removing a 1 mm thick disc of skin from the equatorial section of each fruit. The results were expressed as Kg cm^−2^.

For the extraction of anthocyanins, the protocol of Bongiorni et al. [27] was employed with slight modifications. One gram of flesh fruit was taken from the reddest part of each kiwifruit sample and homogenized in 12 mL of solution composed of methanol/distilled water/hydrochloric acid (36%) in a vol ratio of 50/49/1. The samples were left for 48 h at 4 °C in Teflon-shielded containers in order to prevent the degradation of the extracted anthocyanins. The anthocyanin content was determined by a spectrophotometer (Perkin Elmer Instruments Ltd., Seer Green, Beaconsfield, U.K) reading the absorbance (Abs) of the solution at a wavelength of 535 nm. Subsequently, to calculate the relative concentration of the solution, it was compared with a cyanidin-glucoside standard purchased by Extrasynthese (Genay, France) according to the following formula:y = 0.2251 + 14.873 × Abs (1)

The obtained values were then converted to µg g^−1^ of fresh weight (FW).

### 2.4. AOTF-NIR Spectra Acquisition

A Luminar 5030 miniature portable NIR analyzer (Brimrose Corporation, Baltimore, MD, USA), based on the AOTF-NIR principle, was used for spectral acquisition [19]. This is a portable device that can be used directly in the field, although in this reported experimental trial, the spectral measurements were conducted under laboratory conditions.

Spectra detections were carried out in the range of 1100–2300 nm, with wavelength increments of 2 nm, on each intact fruit through contact between the external gun of the NIR device and the equatorial part of the fruit epicarp, using the diffuse reflectance method of acquisition. Ten fruits for each set of treated (T) and untreated (C) samples at each storage time (from T0 to T4) were spectrally measured by obtaining four hundred spectral readings for each class (*n* = 400), recorded in transmission mode and then averaged for successive chemometric computations.

### 2.5. Chemometric Processing

Spectral performance was tested in the construction of PLS-type regressive models for the predictive evaluation of different destructive parameters under assessment (in detail, anthocyanins and SSC detected on the same fruit, spectrally detected as described before). The spectral data were used as predictor variables (X-block), while the chemical data were used as dependent variables (Y-block). These were then coupled when building up the models. The spectra were preliminarily transformed into absorbance (log 1/T) from the original transmittance units and different pre-treatment filters were tested to identify the best performing ones; in detail, *standard normal variate (SNV)*, Savitzky–Golay filter (1st derivative and 15 points of smoothing), and multiplicative scatter correction (MSC) pre-treatments were tested. The models obtained were cross-validated using the leave-one-out method. The accuracy of the performed PLS models was described by the coefficient of determination in calibration (R^2^ cal) and cross-validation (R^2^ cv), the root mean square error of calibration (RMSEC) and the root mean square error of cross-validation (RMSECV).

### 2.6. Statistical Analyzes

For chemical and technological analyses, the results are expressed as the means ± standard error (SE). Statistical comparisons between different storage times of fruit from different orchard treatments were conducted using the analysis of variance (ANOVA). Tukey’s test, calculated at a 5% significance level, was employed to assess differences between means. Significance was established at a *p*-value < 0.05, and significant differences are indicated by different letters. To ensure comparability and mitigate the effects of varying measurement scales, the data were normalized prior to multivariate analysis using RStudio (R version 4.2.3) [28]. In detail, a principal component analysis (PCA) was performed on the dataset where each row corresponded to an individual sample of red kiwifruit (either T or C) taken at different storage times, while each column represented a specific variable under analysis.

Multivariate analyses for Nir analyses were performed, using Matlab R2013a (MathWorks^®^, Natick, MA, USA) and PLS Toolbox (Eigenvector Research, Inc., Manson, WA, USA).

## 3. Results and Discussion

### 3.1. Cis-3-Hexenyl Butyrate Efficacy on Physical and Chemical Parameters and Anthocyanin Content

As already observed by numerous studies, a gradual increase of WL over the course of storage is generally observed in all types of kiwifruits [9,10,29]. As shown in Table 1, an increase in WL over time for both T and C red kiwifruits has been observed, with no significant differences between samples at each storage time. At the end of storage, we reached about 4–5% of WL. Results are in line with Asadi et al. [30] on red flesh kiwifruit.

In our experiment, both samples showed an increase in SSC (Brix) over time (Table 1). During storage, the C sample exhibited a slower increase in °Brix compared with the T sample, reaching the maximum at the last sampling time (T4) (18.46 ± 0.23), whereas HB-treated samples show their maximum sugar content starting from T2 (18.66 ± 0.32). Obtained results agree with sugar concentrations observed by Cheng et al. [11], who detected, in red kiwifruits, °Brix values ranging from 14.9 to 19.6. Our results are also in line with Asadi et al. [30] after 60 days of storage of different red flesh kiwifruit cultivars (17–20 °Brix of SSC reached).

This suggests that HB treatment has a beneficial effect in maintaining SSC levels during storage, as has already been demonstrated in grape berries, where the fruit of HB-treated plants at 50 mM showed significant differences in sugar contents compared with berries deriving from untreated plants [24].

FFP decreased for all harvested fruit during cold storage (Table 1). Nevertheless, it was observed that the firmness of HB-treated samples decreased significantly starting from T2, according to the SSC data. These highest values demonstrate the efficacy of HB-treatment in accelerating the ripening [22,24,25]. However, at T4, both samples reached the same values, according to data of Asadi et al. [30].

Results underline that HB treatment accelerates the ripening of red kiwifruits, which is evidenced by a quicker increase in SSC and a more rapid softening compared with untreated samples. However, though HB treatment speeds up these processes, the final WL and FFP values of T and C fruits are similar after extended storage. This appears to be aligned with what has been observed in grapefruit storage [24]. Overall, these data show that red kiwifruits treated with HB are characterized by optimal quality attributes for consumption already at T2, compared with C, which has similar values only at T4.

For the color parameters of red kiwifruits, a decrease of L* value in all of the samples was observed throughout the storage, with C samples showing higher L* values than the T ones (Table 2). This difference can be attributed to the HB treatment which, in turn, may have induced a greater intensity of red coloration in the fruit, potentially to the detriment of the L* value. The decrease in the L* parameter observed during the fruit ripening in both samples can be associated with the enhancement of a deeper red hue, basically due to the accumulation of pigments. This observation agrees with the results obtained by Ferrer et al. [31] on the L* values of peach fruit, where peaches of more reddish varieties had shown lower lightness values than yellow peach fruit.

In addition, the h* values of the fruit flesh demonstrate variability across the sampling time points, as can be observed in Table 2. At T0, the treated red kiwifruits exhibit a hue angle of 54.97, which is significantly lower than the untreated fruit equal to 73.6. This difference suggests that the treated fruit may have a different color profile at harvest. Specifically, the lowest h* value recorded for treated samples is 48.09 at T2, indicating a peak in red intensity for treated samples, as observed by Ferrer et al. [31] in peaches, where the h* value decreased during fruit ripening in relation to the increase in red-orange coloration. Obtained data indicate that the treatment influences the h* of red kiwifruits, particularly in the early stages of ripening. In parallel, this could be indicative of slower color development observed in untreated fruit. In fact, it is possible to observe that T red kiwifruits show a* value of 7.32 at the harvest time, indicating a color leaning towards red, while C fruit have a value of 4.54, due to a less red intensity. During ripening, at T2 sampling time, treated fruit reach their maximum a* value at 14.98 (Table 2), signaling a predominance of red color, while untreated fruit drop, confirming the results already observed for h* values.

This parameter was analyzed in red kiwifruits by Peng et al. [32] and it was estimated that during the fruit ripening, the a* value increased.

The data related to the anthocyanin contents (Figure 2) reveal a dynamic change during the storage period in both T and C red kiwifruits. In kiwifruit flesh, anthocyanin concentrations increased progressively during storage in both samples, with significantly higher concentrations in HB-treated red kiwifruits. In T sample the maximum value is reached at T2, while in untreated ones it is observed at T3 and T4. The increase in anthocyanin content at time point T2 suggests not only a competitive advantage for the flesh red color of kiwifruits appreciated by the growers and consumers, but also an obvious nutritional improvement, given the role of these compounds as natural antioxidants. This observation is consistent with previous studies, demonstrating that anthocyanin accumulation in red kiwifruit pulp is absent in fruit that is harvested early, with an average of 4 °Brix. Anthocyanins typically develop concomitantly with the fruit transitions from the color change phase to full ripening [32].

A full and deeper understanding of the effect of HB treatment on red kiwifruit’s quality, a PCA summarizing the differences, as well as relationships between T and C red kiwifruits based on the influence of all of the variables related to their composition (WL, SSC, FFP, L*, h*, a* and anthocyanins) over the different storage times (from T0 to T4) has been computed. The referred results are reported in the graphical biplot included in Figure 3.

The sum of two principal components explains about 90% of the variance in the data (PC1 = 63.9%, and PC2 = 25.7%); original samples, after PCA computation, became the PCA scores, while the analytical variables were derived from the PCA loadings. Reported arrows represent how strongly each variable, displayed as a vector, correlates with these components and the scores segregation; the direction and length of these vectors indicate their discriminative influence and impact. The positioning of the T and C samples in the biplot shows the effect of the treatment on the composition of the red kiwifruits. Based on the separation of samples over the storage times, it can be observed that WL increases over time for both samples, as indicated by the partially positive contribution to both PC1 and PC2. Meanwhile, FFP decreases over time by negatively affecting PC1. This is a combined observation that, as storage time progresses, T and C kiwifruits lose more weight and become softer, as expected. These trends are typically found during storage as a result of water evaporation, even if only in a limited manner when under controlled conditions, and the fruit’s cell wall structure breaking down, leading to a firmness reduction [9]. Regarding color data, L* shows a negative contribution to PC1, suggesting that, as the storage time increases, the L* values decrease. This indicates a general loss of lightness in the red kiwifruits over time for both samples. Hue contributes positively to PC2. The positioning of the T and C samples associated with the hue vector indicates differences in color development over time. Meanwhile, the anthocyanin vector likely correlates positively with the hue vector. As anthocyanin levels increase, the hue value becomes more intense, enhancing the overall color quality of the red kiwifruits. Anthocyanins and a* value show positive associations with PC1, and are also positively correlated between them. The positioning of the T and C samples along the a* vector suggests how the red coloration changes over the storage period, indicating that the treatment can have an effect in enhancing the red color intensity of the fruit, also associated with a progressive increase of anthocyanin levels as the fruit ripens. The interaction between the a* and SSC vectors in the biplot indicates that, as the kiwifruits ripen, there is a simultaneous enhancement in both red color intensity and sweetness, usually observed as an effect of ripening enhancement [30]. The SSC vector contributes positively to PC1, confirming that, as storage time progresses, SSC in both samples tends to increase. In conclusion, the biplot shows that both treatment and storage time significantly influence the physical and biochemical properties of red kiwifruits, also revealing critical changes in WL, firmness, color intensity, and sweetness. These insights emphasize the potential of treatment to enhance the overall quality of the red kiwifruits over time.

### 3.2. Nir Spectral Acquisition Models for Anthocyanins and SSC

In parallel to the destructive evaluation of kiwifruits, NIR spectroscopy has been tested as a non-destructive and predictive tool addressed to fruit quality detection. In Figure 4, the averaged absorbance spectra (1st derivative) derived from all of the spectral kiwifruit measurements are shown; graphical representation is useful in defining the main vibrational behavior of the fruit tissue, associated with its molecular content. The first band of spectral significance observed at 1150 nm is generally assigned to the second overtone of the combination of the symmetric and asymmetric OH stretching and bending bands. Additionally, the band intercepted at around 1750 nm, which exhibits relatively low absorbance, corresponds to the first overtone of the CH stretching vibrations in CH_3_, CH_2_, and CH=CH groups [33,34].

The last significant band detected in the range of 2200–2260 nm refers to the resonances attributed to the combination of C-H stretching and CH_2_ group in vibrational deformation [35].

As typical for vegetal matrices that are spectrally measured, the absorbance spectra are characterized by two main water absorption bands at approximately 1450 nm and 1920–1950 nm [34]. These are assigned to the first tone of the symmetrical and asymmetrical OH stretching and/or combination bands (1450 nm), as well as to the combination of the OH stretching band and OH bending band (1920–1950 nm) related to the water molecules [33,36].

The same detected spectra were used for the building up of regressive models performed by PLS chemometric approach aimed at predicting anthocyanin content in whole kiwifruits. Spectra were transformed into absorbance (log 1/T) and, in turn, treated with various chemometric filters to identify the most performant of the regressive computations. In Table 3 the statistical indexes referring to the model performance for anthocyanin content are reported.

The best result was obtained by applying a *standard normal variate (SNV)* filtering, before PLS computation. This is a very efficient statistical filtering when reducing the within-class variance, as described by Barnes et al. [37]. The obtained model shows coefficients of determination of 0.986 and 0.860 in calibration (R^2^ cal) and cross-validation (R^2^ cv), respectively. The obtained RMSEC is equal to 1.60, and the RMSECV to 4.83 in calibration (RMSEC and cross-validation, respectively) while seven latent variables (LVs) minimize the predicting error (Table 3). Predictive performances of PLS models was assessed by the leave-one-out cross-validation method. This method is considered appropriate when a limited dataset of samples is addressed to the regressive computation. The ratio of the root means square error in prediction (RMSECV) to the dimensionality of the destructive data (SD) resulted in a ratio of performance to deviation (RPD) of 2.83. This value was found to be above the limit of good predictive performance, which some authors have estimated at 2.5, so the optimal value of the index was instead assigned as 3 [38,39].

In the literature, several authors have investigated the ability of vibrational spectroscopy, in combination with chemometric statistical techniques, aimed at predicting anthocyanin content in various vegetal crops [18,19,40].

However, until now, no one has investigated this application specifically in kiwifruit, even though the aptitude of NIRs has already been tested on this fruit in order to detect other quality attributes, like firmness, soluble solids, and dry matter [41]. With respect to anthocyanins, although it is widely known that their absorption response is stronger in correspondence with the visible range of light spectra, significant correlations for predicting the content of these pigments have also been found in the NIR region of the electromagnetic spectrum as reported in the available literature [42,43]. Due to the complex interplay of absorption mechanisms and overlapping bands, establishing a direct link between the concentration of anthocyanins, but more generally of phenolic compounds, and the dimensionality of these bands is still a challenge [44].

Mariani et al. [45], working on the prediction of the total anthocyanin content with NIRs on jaboticaba fruit, identified the spectral regions between 1232–1279 nm, 1320–1522 nm, 1792–2009 nm as the largest contributors of the spectra to regression modeling. In the present study, in terms of the spectra contribution to PLS modeling, correlation peaks were found in the region between 1156–1180 nm, at 1343 nm, and at 1888 nm.

On the other hand, significant spectra contribution to PLS modeling for SSC detection was found in correspondence with the regions between 1100–1250 nm and 1850–1950 nm, in line with what was reported by other authors. In a recent study, Cevoli et al. [16], working with an FT-NIR device on the variety *A. chinensis (Jintao)*, found a spectral significance in the region of the spectrum between 1850 and 2000 nm. According to McGlone & Kawano [46], these portions of the light spectrum are attributed to the vibrational response of the glucidic component of the matrix.

In Table 4, the results in terms of model performance for SSC are reported. The best results were achieved by model computation where spectral information was transformed into absorbance, and no chemometric filters were applied. The performed model shows coefficients of determination of 0.888 and 0.784 in calibration and cross-validation, respectively. The mean square error is of 0.77 and 1.07 in calibration (R^2^ c) and cross-validation (R^2^ cv), respectively, while six latent variables (LVs) minimize the predicting error. RPD is equal to 2.19, just below the value of good predictive performance. Generally speaking, to calibrate and validate NIR as a viable method for predicting soluble solids content in fruit, a wide range of work is required to ensure that the calibration is robust [34,39]. In the experimental trial here proposed, the spread of data referring to the soluble solids content varies between 11.30 and 19.30 °Brix; however, 60% of cases fall in the range of 17–19 °Brix. The results are therefore not optimally distributed, and the model is calibrated using a narrower range than if the data were normally distributed between the minimum and maximum values. This distribution, however, influences the RPD value, resulting in a lower index result; for more robust and significant results a larger dataset for both spectra and measured SSC is desirable. Nevertheless, performed results indicate a very good perspective, and they confirm the feasibility of the NIR technique for the goal of estimating quality attributes in red kiwifruits, considering further in-depth analyses and model improvement.

## 4. Conclusions

In conclusion, the data from the study on red kiwifruits treated with HB reveal several important correlations between WL, sugar content (°Brix), firmness, color parameters (L*, a*, h), and anthocyanin accumulation during storage. These parameters are all closely linked to the ripening process and demonstrate that the HB treatment significantly impacts the fruit’s ripening dynamics, possibly accelerating the process compared with untreated fruit. Color changes observed during storage provide further insights into the effect of HB treatment. Treated fruit showed lower L* values and lower h* values, correlating with higher a* values, which signify a more intense red coloration. This suggests that HB treatment may accelerate the biosynthesis of anthocyanins, also enhancing the visual appeal of the fruit earlier in storage.

Moreover, the obtained results highlight the feasibility of NIR technology in discriminating the presence and concentration of anthocyanins and SSC. The combined insights from HB treatment and NIR spectroscopy underscore their potential for improving kiwifruit quality, offering a background for further studies and practical applications in commercial settings. In conclusion, this preliminary study suggests the promising effect of HB treatment on the ripening quality parameters of red kiwifruit (namely on anthocyanins). With the goal to confirm the obtained data, to better elucidate the treatment effects on the plant’s defense mechanisms and fruit ripening, and finally to explore a possible commercial application, we are considering a future investigation anticipating the timing of the in-field plant’s treatments at lower HB concentrations.

## Figures and Tables

**Figure 1 foods-14-00480-f001:**
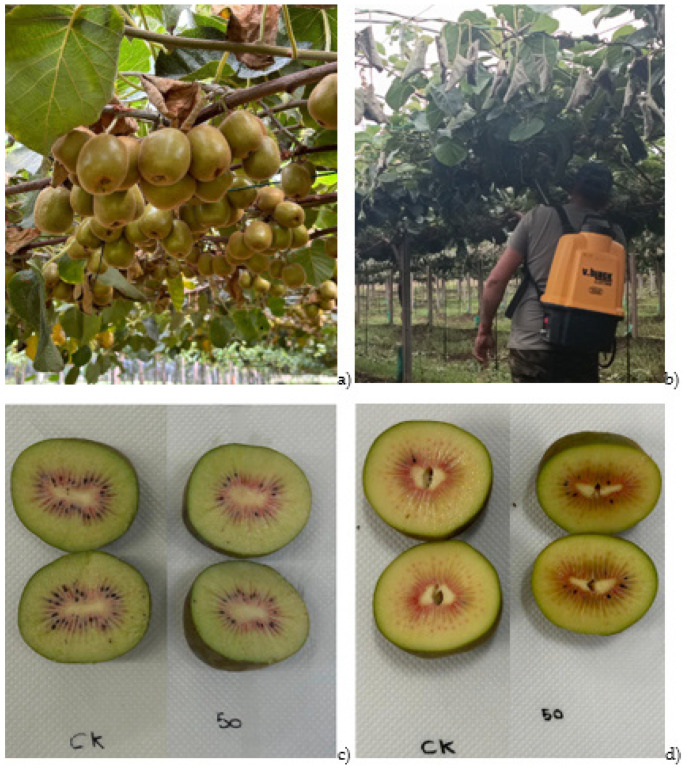
Experimental trial: (**a**) red kiwifruit plants, (**b**) red kiwifruit plants’ spray treatment with cis-3-hexenyl butyrate (HB), (**c**) untreated (C; CK) and treated (T; 50) red kiwifruits at 0 days (T0) of cold storage at 1 °C, and (**d**) untreated (C; CK) and treated (T; 50) red kiwifruits at 30 days (T2) of cold storage at 1 °C.

**Figure 2 foods-14-00480-f002:**
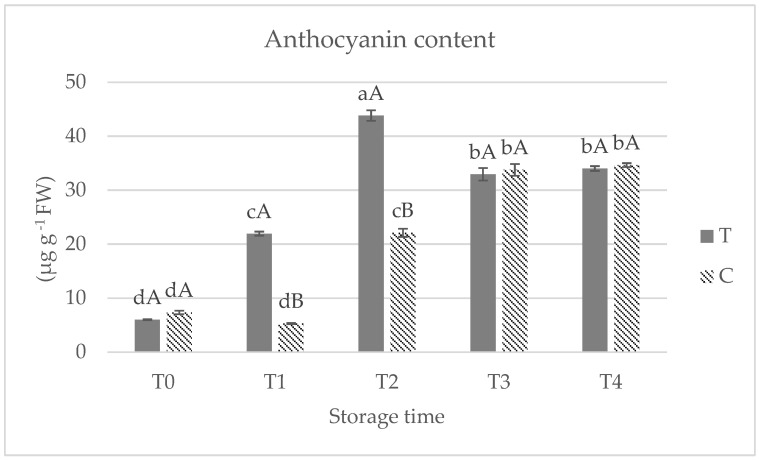
Anthocyanin content in treated (T) and untreated (C) red flesh kiwifruits at 0 days (T0), 15 days (T1), 30 days (T2), 45 days (T3) and 60 days (T4) of cold storage at 1 °C. Capital letters indicate comparisons between different treatments of fruit at each specific time. Lowercase letters reflect comparisons between different storage times for each sample fruit of the same treatment. According to the Tukey test (*p* < 0.05), there are no significant differences between means that share the same letter.

**Figure 3 foods-14-00480-f003:**
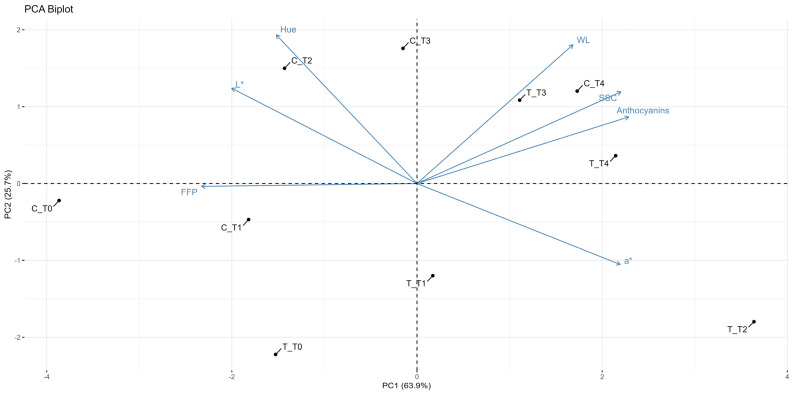
Biplot (PC1 vs. PC2) of PCA built with physical and chemical parameters (WL, weight loss; SSC, soluble solid content; FFP, flesh firmness penetrometer; L*, lightness; Hue, hue angle; a*, green–red axis; anthocyanins, anthocyanin content) for treated (T) and untreated (C) red kiwifruit samples during storage: T0 (0 days), T1 (15 days), T2 (30 days), T3 (45 days) and T4 (60 days).

**Figure 4 foods-14-00480-f004:**
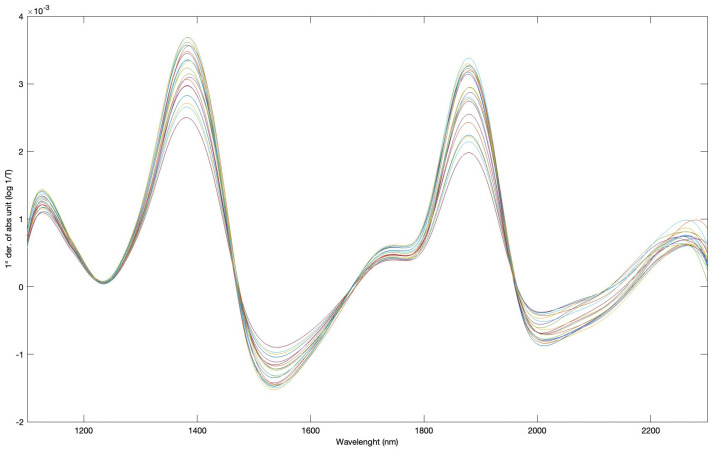
Averaged absorbance spectra (log 1/T), in the 1st derivative, obtained from AOTF-NIR acquisitions. The average spectra of kiwis belonging to various sample groups are presented. The peaks with the highest absorbance correspond to the vibrational character of the resonant functional groups.

**Table 1 foods-14-00480-t001:** Physicochemical parameters (WL, weight loss; SSC, soluble solid content; FFP, flesh firmness penetrometer) of treated (T) and untreated (C) red kiwifruits at 0 days (T0), 15 days (T1), 30 days (T2), 45 days (T3) and 60 days (T4) of cold storage at 1 °C.

Parameter	Treatments	T0	T1	T2	T3	T4
WL (%)	T	-	1.01 ± 0.09 (cdA)	2.73 ± 0.40 (cdbA)	3.22 ± 0.20 (abA)	4.06 ± 0.41 (aA)
	C	-	1.79 ± 0.44 (dA)	2.7 ± 0.41 (bcdA)	4.39 ± 0.60 (abA)	4.92 ± 0.39 (aA)
SSC (Brix)	T	13.38 ± 0.74 (deA)	15.80 ± 0.70 (bcA)	18.66 ± 0.32 (aA)	18.70 ± 0.15 (aA)	18.80 ± 0.33 (aA)
	C	12.06 ± 0.56 (eA)	14.52 ± 0.27 (cdA)	17.58 ± 0.14 (abB)	17.38 ± 0.14 (abB)	18.46 ± 0.23 (aA)
FFP (Kg cm^−2^)	T	7.12 ± 0.30 (cB)	5.22 ± 0.37 (dB)	3.08 ± 0.12 (deB)	2.47 ± 0.50 (eB)	2.12 ± 0.30 (eA)
	C	10.95 ± 0.42 (aA)	8.21 ± 0.18 (abA)	6.04 ± 0.17 (bA)	4.07 ± 0.55 (bcA)	2.13 ± 0.23 (eA)

Note: Capital letters indicate comparisons between different treatments of fruit at each specific time. Lowercase letters reflect comparisons between different storage times for each sample fruit of the same treatment. According to the Tukey test (*p* < 0.05), there are no significant differences between means that share the same letter within the same column or row.

**Table 2 foods-14-00480-t002:** Color parameters (L*, lightness; h*, hue angle; a*, Green-Red axis) of treated (T) and untreated (C) red kiwifruits at 0 days (T0), 15 days (T1), 30 days (T2), 45 days (T3) and 60 days (T4) of cold storage at 1 °C.

Parameter.	Treatments	T0	T1	T2	T3	T4
L*	T	54.66 ± 1.82 (bcB)	50.32 ± 2.32 (dB)	42.61 ± 0.82 (efB)	45.53 ± 1.45 (eB)	45.55 ± 1.60 (eB)
	C	65.20 ± 1.50 (aA)	58.85 ± 1.78 (abcA)	52.13 ± 1.80 (bcA)	52.06 ± 0.6 (bcA)	52.90 ± 0.39 (bcA)
h*	T	54.98 ± 2.96 (bcB)	62.66 ± 4.05 (abA)	48.09 ± 2.22 (cB)	70.45 ± 1.90 (aA)	62.84 ± 4.27 (abA)
	C	73.60 ± 3.44 (aA)	70.63 ± 1.20 (aA)	74.19 ± 2.04 (aA)	71.06 ± 1.70 (aA)	65.89 ± 2.96 (abA)
a*	T	7.32 ± 0.71 (bcdA)	9.79 ± 0.43 (bA)	14.98 ± 0.56 (aA)	8.86 ± 0.23 (bA)	9.27 ± 0.77 (bA)
	C	4.55 ± 1.30 (cdB)	6.46 ± 1.24 (bcdA)	3.80 ± 1.11 (dB)	7.97 ± 0.21 (bA)	8.39 ± 0.26 (bA)

Note: Capital letters indicate comparisons between different treatments of fruit at each specific time. Lowercase letters reflect comparisons between different storage times for each sample fruit of the same treatment. According to the Tukey test (*p* < 0.05), there are no significant differences between means that share the same letter within the same column or row.

**Table 3 foods-14-00480-t003:** Modeling performances in anthocyanin content prediction resulting from different pre-treatments applied to NIR spectra measurements. Indexes include R^2^ (coefficient of determination) for calibration (R^2^_cal) and cross-validation (R^2^_cv), root mean square error of calibration (RMSEC), root mean square error of cross-validation (RMSECV), the number of latent variables (LV), and the ratio of performance to deviation (RPD).

Pre-Treatment	R^2^ cal	R^2^ cv	RMSEC	RMSECV	LV	RPD
-	0.964	0.594	2.5452	8.7862	6	1.56
ABS	0.868	0.754	4.8413	6.6595	4	2.05
SNV	0.986	0.871	1.5969	4.8304	7	2.83
SG derivatives 1^	0.986	0.860	1.5636	5.0582	5	2.70
MSC + autoscale	0.953	0.803	2.8866	5.9376	5	2.30

**Table 4 foods-14-00480-t004:** Modeling performances in total soluble solids content (SSC) detection resulting from different pre-treatments applied to NIR spectra detections correlated with total soluble solids content (SSC). Indexes include coefficient of determination (R^2^) in calibration (R^2^_cal) and cross-validation (R^2^_cv), root mean square error of calibration (RMSEC), root mean square error of cross-validation (RMSECV), the number of latent variables (LV), and the ratio of performance to deviation (RPD).

Pre-Treatment	R^2^ cal	R^2^ cv	RMSEC	RMSECV	LV	RPD
-	0.859	0.692	0.8616	0.2966	5	1.81
ABS	0.888	0.784	0.7671	1.0746	6	2.19
SNV	0.872	0.776	0.8219	1.0939	5	2.15
SG derivatives 1^	0.914	0.740	0.6707	1.1932	4	1.97
MSC + autoscale	0.941	0.713	0.555	1.278	8	1.84

## Data Availability

Data are contained within the article.

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
