# Peer review of "Destructive and Non-Destructive Evaluation of Anthocyanin Content and Quality Attributes in Red Kiwifruit Subjected to Plant Spray Treatment with Cis-3-Hexenyl Butyrate"

_foods, 2025, doi:10.3390/foods14030480_

Round 1
Reviewer 1 Report
Comments and Suggestions for Authors
This is an interesting piece of work that also has significant practical value, as it has discovered that HB has a good effect on enhancing the red color of fruits. However, there are several points in the manuscript that need to be revised and improved:
1. Line 107-108, Eight treated plants (T) and eight control plants (C) were selected on a sample basis within the row. How many harvested fruits used in this study totally?
2. In Line 123-124, it said: At T0 and during each sampling time, twenty-five fruits of untreated and treated samples were weighed using a digital balance to monitor WL (%) during cold storage. However, when measuring SSC and anthocyanin, it was not indicated how many fruits the samples were taken from.
3. Line 270 Figure1 shows that T sample the maximum value is reached at T2, while in untreated ones is observed at T3 and T4. However, the commercial value of the treatment’s impact on the fruit’s anthocyanin content was not elucidated. At the harvest period T0, there was no difference in anthocyanin content between the treated and control fruits, but at T1 and T2, the anthocyanin content in the treated fruits was significantly higher than that in the control, with no difference observed at T3 and T4 stages.
4. This work evaluates red kiwifruit plants' spray treatment with cis-3-hexenyl bu-tyrate (HB) as an inductor of some metabolic mechanisms related to fruit ripening, please provide more explanation on the safety assessment of cis-3-hexenyl butyrate (HB).
Author Response
January 25, 2025
Dear Reviewer
I submit the revised text of the manuscript "Destructive and non-destructive evaluation of anthocyanin content and quality attributes in red kiwifruit subjected to plant spray treatment with cis-3-hexenyl butyrate" according to the observations.
As was requested, a file for each reviewer was included listing the point-by-point responses to the specific formalized questions.
The amendments in the manuscript were made using a track change form in red font.
The authors acknowledge the Reviewers for providing their suggestions which contribute to the overall improvement of the work.
Best regards,
Rinaldo Botondi
rbotondi@unitus.it
Answers
Reviewer 1
- Line 107-108, Eight treated plants (T) and eight control plants (C) were selected on a sample basis within the row. How many harvested fruits used in this study totally?
- Done in the text. Lines 108-109.
- In Line 123-124, it said: At T0 and during each sampling time, twenty-five fruits of untreated and treated samples were weighed using a digital balance to monitor WL (%) during cold storage. However, when measuring SSC and anthocyanin, it was not indicated how many fruits the samples were taken from.
- Done in the text. Lines 134-137.
- Line 270 Figure1 shows that T sample the maximum value is reached at T2, while in untreated ones is observed at T3 and T4. However, the commercial value of the treatment’s impact on the fruit’s anthocyanin content was not elucidated. At the harvest period T0, there was no difference in anthocyanin content between the treated and control fruits, but at T1 and T2, the anthocyanin content in the treated fruits was significantly higher than that in the control, with no difference observed at T3 and T4 stages.
- Done in the text. Lines 280-283.
-
- This work evaluates red kiwifruit plants' spray treatment with cis-3-hexenyl bu-tyrate (HB) as an inductor of some metabolic mechanisms related to fruit ripening, please provide more explanation on the safety assessment of cis-3-hexenyl butyrate (HB).
- cis-3-hexenyl butyrate (HB) is an organic compound found in the volatile profile of numerous fruit products such as: passion fruit, plum, strawberry, apricot, lemon, etc. (VCF (Volatile Compounds in Food): Database/Nijssen, L.M.; IngenVisscher, C.A. van; Donders, J.J.H. (eds). - Version 15.1 - Zeist (The Netherlands): TNO Triskelion, 1963-2014). Toxicology Classification: Cramer Classification - Class I, Low.

Reviewer 2 Report
Comments and Suggestions for Authors
This manuscript studied the destructive and non-destructive evaluation of anthocyanin content and quality attributes in red kiwifruit subjected to plant spray treatment with cis-3-hexenyl butyrate. This is interesting, and build a link between pre- and post-harvest of red kiwifruit, which is very useful for growers and researchers in this field. The results are solid and instructive, and language are clear to read. However, some detailed points in the manuscript need to be modified.
1-Please provide some photos, including red kiwifruit tree and fruit from TR\CK; appearance of fruit in storage, either photos in the procedure or at the end of storage. In which, readers can see the grow conditions, fruit appearance, decay and so on.
2- Please provide the data of fruit yield or average weight of fruit in two treatments, to detect whether treatment influence growth of red kiwifruit tree.
3-In this paper, authors used two methods: destructive and non-destructive evaluation, what about the accuracy of them, which one is more accuracy and believable, and how to prove it?
4-How to predict the anthocyanin and SSC content concluded in this paper? What about the accuracy of this method? Lack of necessary and enough data and subsequent prove.
5-fruits modified to fruit.
6-Line 118, Relative Humidity modified to relative humidity
7-Line 145, modify the format of equation according to the journal guidelines.
8-Introduction, generally no more than 4 paragraphs better.
9-Conslusion must be shortened or modified to Conslusion and Prospective.
Author Response
January 25, 2025
Dear Reviewer
I submit the revised text of the manuscript "Destructive and non-destructive evaluation of anthocyanin content and quality attributes in red kiwifruit subjected to plant spray treatment with cis-3-hexenyl butyrate" according to the observations.
As was requested, a file for each reviewer was included listing the point-by-point responses to the specific formalized questions.
The amendments in the manuscript were made using a track change form in red font.
The authors acknowledge the Reviewers for providing their suggestions which contribute to the overall improvement of the work.
Best regards,
Rinaldo Botondi
rbotondi@unitus.it
Answers to Reviewer2:
1-Please provide some photos, including red kiwifruit tree and fruit from TR\CK; appearance of fruit in storage, either photos in the procedure or at the end of storage. In which, readers can see the grow conditions, fruit appearance, decay and so on.
- Thank you for your suggestion. Done in the text. New Figure 1. Lines: 112-119.
2- Please provide the data of fruit yield or average weight of fruit in two treatments, to detect whether treatment influence growth of red kiwifruit tree.
2.Done in the text. Lines: 121-122.
3-In this paper, authors used two methods: destructive and non-destructive evaluation, what about the accuracy of them, which one is more accuracy and believable, and how to prove it?
3. The accuracy of destructive and non-destructive (NIR spectroscopy) analytical procedures are validated by the statistical methods of significance used in the conducted experiments. In particular, regarding the use of NIR, non-destructive analyses of SSCs have been extensively documented and validated in several fruits, whereas for anthocyanins, the results are new and original in this research.
4-How to predict the anthocyanin and SSC content concluded in this paper? What about the accuracy of this method? Lack of necessary and enough data and subsequent prove.
4. Agree with your observation. The data on SSC and anthocyanin content was only necessarily derived from the last season (2023) of HB treatment on kiwifruit plants. In the recent season (2024), we have repeated the experimental trial with more treatments, anticipating their timing and using different HB concentrations. This aims to validate the data with increased soundness.
5-fruits modified to fruit.
5. Done in the text.
6-Line 118, Relative Humidity modified to relative humidity.
6. Done in the text. Line 127.
7-Line 145, modify the format of equation according to the journal guidelines.
7. Done in the text Lines: 157-158.
8-Introduction, generally no more than 4 paragraphs better.
8.Thank you for the suggestion. However, we believe that the content is already well-balanced, providing the right level of detail. This allows us to keep the focus on the main points without losing clarity.

Reviewer 3 Report
Comments and Suggestions for Authors
In this study, the quality changes of red kiwifruit treated with cis-3-Hexenyl Butyrate (HB) during storage were investigated and the ripening process of red kiwifruit was monitored by Near-Infrared (NIF) spectrum. The results will provide a theoretical basis for the application of HB in enhancing the quality of red kiwifruit and the establishment of a non-destructive fruit ripening monitoring method. However, there are some several scientific concerns about this manuscript.
1. Line 27, It is recommended to reserve 4 to 5 keywords.
2. Line 99, It is suggested to add sensory evaluation experiment to make the results more convincing.
3. Line 209, Does faster ripening mean the fruit is more perishable and has a shorter shelf life?
4. Lines 216-218, What is the best quality standard for red kiwifruit ripeness? Is T2 up to the standard?
5. Table 1, Please make sure the data in the table has the same number of decimal places.
6. Figure 1, The figure shows that anthocyanin content shows a decreasing trend after T2. May I ask whether HB has advantages in application?
7. Lines 266-269, What is the mechanism of HB affecting the change of anthocyanin content? Does HB only work during the ripening process after harvest?
8. Line 322, Why is the NIF model of only two indexes established?
9. Tables 3 and 4, Both models need to be verified experimentally.
Author Response
January 25, 2025
Dear Reviewer
I submit the revised text of the manuscript "Destructive and non-destructive evaluation of anthocyanin content and quality attributes in red kiwifruit subjected to plant spray treatment with cis-3-hexenyl butyrate" according to the observations.
As was requested, a file for each reviewer was included listing the point-by-point responses to the specific formalized questions.
The amendments in the manuscript were made using a track change form in red font.
The authors acknowledge the Reviewers for providing their suggestions which contribute to the overall improvement of the work.
Best regards,
Rinaldo Botondi
rbotondi@unitus.it
Answers to Reviewer3
1. Line 27, It is recommended to reserve 4 to 5 keywords.
A: We delete two keywords. Lines: 26-27.
- Line 99, It is suggested to add sensory evaluation experiment to make the results more convincing.
A: Thank you for your suggestion. Sensory evaluation is certainly a key subject to understanding the overall quality of red kiwifruits. We will consider this aspect in future trials.
- Line 209, Does faster ripening mean the fruit is more perishable and has a shorter shelf life?
A: HB-treatment accelerates the ripening of red kiwifruits starting from T2, but the investigated quality parameters (FFP and SSC) have the same trend of the control samples, with no differences at the end of the storage (T4). So, the period of shelf life is the same between the two different samples.
- Lines 216-218, What is the best quality standard for red kiwifruit ripeness? Is T2 up to the standard?
A: Results in Table 1 show that the reached values of the considered parameters (WL, FFP, and SSC) at T2 for the T sample suggest that the product is ready to be consumed.
- Table 1, Please make sure the data in the table has the same number of decimal places.
A: Done in Table 1. Thank you. Lines: 234-239.
- Figure 1, The figure shows that anthocyanin content shows a decreasing trend after T2. May I ask whether HB has advantages in application?
A: The advantage of using HB treatments is to anticipate the marketability of red kiwifruits reaching the optimum overall quality 15 days before the untreated sample.
- Lines 266-269, What is the mechanism of HB affecting the change of anthocyanin content? Does HB only work during the ripening process after harvest?
A: Bibliographic studies (references number 23, 24, and 25) suggest that HB treatment has an effect on the defense mechanisms of the plant with consequent closure of the stomata and accumulation of the anthocyanin content in the fruit following the induced stress. As explained in the conclusions, future investigation of the effects of the treatment in the field will allow us to fully understand the effects of the HB treatment on red kiwifruit plants.
- Line 322, Why is the NIF model of only two indexes established?
A: The two selected parameters for developing the NIR models have been identified with the goal to: a) test the non-destructive technique in predicting the ripening stage of the kiwifruits (SSC); b) evaluating the NIR aptitude in correlating with anthocyanins which are the metabolites suggested to be associated with in-plant treatment. In any case, as reported in the paper, the two indexes are included into a large number of chemical and physical parameters well-known to be combined with NIR spectroscopy in kiwifruits and in fruits and vegetal in general.
9.Tables 3 and 4, Both models need to be verified experimentally.
A: NIR spectroscopy, as well as all the non-destructive techniques sensor-based, are supported by modeling approaches based on multivariate statistics. In those approaches both calibrating and predicting results were tested and they are considered the experimental verification of the device performances; as it is possible to find in the MS, all the statistical indexes of performances for carried out models are reported and discussed in their significance. Considering the limited number of samples used, a larger dataset would help confirm the models' reliability and improve their future applicability, but all these considerations were well reported and clarified in the paper.

Reviewer 4 Report
Comments and Suggestions for Authors
The study investigated the effect of cis-3-hexenyl butyrate pre-harvest treatment on kiwifruit anthocyanins and the efficacy of NIR to estimate the quality of its parameters. Developing non-destructive models to avoid wet chemistry in future is the modern way of conducting trials in the post-harvest physiology of fresh fruit. The study is interesting, but the manuscript can be further improved following my specific comments below:
Title:
- Can be clarified along the lines of 'Investigating the effect of cis-3-Hexenyl Butyrate preharvest spray on kiwifruit quality and the quality predictability using NIR regression models'
- If the colour is mentioned, just mention the entire cultivar name i.e 'Red Sun'
Abstract:
L10: evaluates should be in past tense.
L17: 'NIR spectral acquisition was tested' sounds incorrect. The spectra was acquired but the acquisition was not tested, rather its ability to estimate anthocyanins and SSC.
L24: Instead of 'predicting' implying a random value, 'estimation' can be used
Introduction:
L30: mention the scientific name of kiwifruit.
L38: The scientific name cannot be abbreviated on its first mention.
L45: Use a synonym of frame.
L56: Affecting the ease of processing.
L68-L69: It is arguably.
L69: ...of Actinidia [16]. I suggest use of a common name, and mentioning the scientific name once. The paper is going to be read by people who don't work on fruits and they should not be lost in meaning.
L72: Scientific names are always written in italics
L82-L85: This sentence can be broken into two for clarity
Material and Methods:
L105: What time is the early morning hours? i.e 06h00 to 08h00 or 07h00 to 11h00
L123: 25 treated and '25' untreated. Otherwise, this line means the 25 fruit are divided into two groups
L136: Slight modifications, appropriate means the initial methodology was incorrect.
L149: not detection, acquisition.
L150: ...although in the here reported... grammar
L153: Where on the fruit was the spectra collected? mention it like you did in line 132
L175: Standard error (se)
Results and Discussion:
L189: It is normally better to remind the reader what the abbreviation mean in this section. Avoid beginning with HB, rather use full word.
L213: Brix is a unit of SCC. There was an increase in SCC.
L353: The title can be improved for clearance. Model parameters? spectral data? anthocyanins concentration?
L361: R2
L362: R2
L362: 1.5969 is 1.60
L377: ...with respect to anthocyanins,
L387 and L388: is it supposed to be 'Modeling' or 'Modelling'?
L397: in line with what reported other authors. Grammar.
L398: Scientific names are always in italics.
L402: In Table 4 results... there is a grammatical error here.
L406: 0.76 and 1.07 are incorrect. double-check.
L407: Is it 6 or 7 LVs?
L419: estimate rather than predict.
Conclusion:
The only conclusion lines necessary are L430-L432 and L450-L456. The rest is repetition of results.
Author Response
January 25, 2025
Dear Reviewer
I submit the revised text of the manuscript "Destructive and non-destructive evaluation of anthocyanin content and quality attributes in red kiwifruit subjected to plant spray treatment with cis-3-hexenyl butyrate" according to the observations.
As was requested, a file for each reviewer was included listing the point-by-point responses to the specific formalized questions.
The amendments in the manuscript were made using a track change form in red font.
The authors acknowledge the Reviewers for providing their suggestions which contribute to the overall improvement of the work.
Best regards,
Rinaldo Botondi
rbotondi@unitus.it
Answers to Reviewer4
1. Can be clarified along the lines of 'Investigating the effect of cis-3-Hexenyl Butyrate preharvest spray on kiwifruit quality and the quality predictability using NIR regression models'
A: Thank you for your suggestion, but the adopted title highlights the correlation between destructive and non-destructive analysis of the red kiwifruit quality.
- If the colour is mentioned, just mention the entire cultivar name i.e 'Red Sun'
A: The designated cultivar BAG1 has yet to be registered and therefore cannot be mentioned in the title.
- L10: evaluates should be in past tense.
A: Done in the text. Line 10.
- L17: 'NIR spectral acquisition was tested' sounds incorrect. The spectra was acquired but the acquisition was not tested, rather its ability to estimate anthocyanins and SSC.
A: Performed. Line 17. Thank you.
- L24: Instead of 'predicting' implying a random value, 'estimation' can be used.
A: Done in the text. Line 24.
- L30: mention the scientific name of kiwifruit.
A: Done in the text. Line 30.
- L38: The scientific name cannot be abbreviated on its first mention.
A: Thank you, but we added it in line 30.
- L45: Use a synonym of frame.
A: Done in the text. Line 45.
- L56: Affecting the ease of processing.
A: Done in the text. Line 56.
- L68-L69: It is arguably.
A: Done in the text. Line 68.
- L69: ...of Actinidia [16]. I suggest use of a common name, and mentioning the scientific name once. The paper is going to be read by people who don't work on fruits and they should not be lost in meaning.
A: Done in the text. Line 69.
- L72: Scientific names are always written in italics.
A: Done in the text. Line 72.
- L82-L85: This sentence can be broken into two for clarity.
A: Done in the text. Line 83.
- L105: What time is the early morning hours? i.e 06h00 to 08h00 or 07h00 to 11h00.
A: early morning hours (starting from 07:00 a.m.). Done in the text. Line 104.
- L123: 25 treated and '25' untreated. Otherwise, this line means the 25 fruit are divided into two groups
A: Done in the text. Lines: 132-133.
- L136: Slight modifications, appropriate means the initial methodology was incorrect.
A: Done in the text. Line 148.
- L149: not detection, acquisition.
A: Done in the text. Line 162.
- L150: ...although in the here reported... grammar
A: Corrected. Line 163.
- L153: Where on the fruit was the spectra collected? mention it like you did in line 132
A: The spectra were collected from the equatorial region of the whole fruits. Amended in the text. Line 168
- L175: Standard error (se).
A: Done in the text. Line 189.
- L189: It is normally better to remind the reader what the abbreviation mean in this section. Avoid beginning with HB, rather use full word.
A: Done in the text. Line 203.
- L213: Brix is a unit of SCC. There was an increase in SCC.
A: Done in the text. Line 228.
- L353: The title can be improved for clearance. Model parameters? spectral data? anthocyanins concentration?
A: Thanks for your suggestion. We apported some modifications to captions of table 3 and 4 respectively, hoping to have improved their clarity as you requested.
- L361: R2
A: Done in the text. Lines 185, 378, and 422-423.
- L362: R2
A: Done in the text. Lines 185, 378, and 422-423.
- L362: 1.5969 is 1.60
A: Done in the text. Line 379.
- L377: ...with respect to anthocyanins,
A: Done in the text. Line 393.
- L387 and L388: is it supposed to be 'Modeling' or 'Modelling'?
A: modeling. Thank you.
- L397: in line with what reported other authors. Grammar.
A: Done in the text. Line 413.
- L398: Scientific names are always in italics.
A: Done in the text. Line 414.
- L402: In Table 4 results... there is a grammatical error here.
A: Done in the text. Line 418.
- L406: 0.76 and 1.07 are incorrect. double-check.
A: 0.77 and 1.07. Done in the text. Line 416.
- L407: Is it 6 or 7 LVs?
A: 6. Done in the text. Line 422.
- L419: estimate rather than predict.
A: Done in the text. Line 434.
- The only conclusion lines necessary are L430-L432 and L450-L456. The rest is repetition of results.
A: Done in the text. Lines: 438-457.

Round 2
Reviewer 2 Report
Comments and Suggestions for Authors
Authors have replied all the concerns, and the paper can be accepted.
Reviewer 4 Report
Comments and Suggestions for Authors
Thank you for addressing my comments. I suggest the English editing is done to finalize the manuscript.
Comments on the Quality of English Language
The authors English is not at the level of publication yet. But, the concept and report is well-presented. I suggest the English editing and then publication of the manuscript.